# Biaxial Flexural Strength and Vickers Hardness of 3D-Printed and Milled 5Y Partially Stabilized Zirconia

**DOI:** 10.3390/jfb16010036

**Published:** 2025-01-20

**Authors:** Sebastian Hetzler, Carina Hinzen, Stefan Rues, Clemens Schmitt, Peter Rammelsberg, Andreas Zenthöfer

**Affiliations:** Department of Prosthodontics, Medical Faculty, Heidelberg University, Im Neuenheimer Feld 400, 69120 Heidelberg, Germany

**Keywords:** 3D printing, additive manufacturing, biaxial flexural strength, Vickers hardness, translucent, 5Y-stabilized zirconia

## Abstract

This study compares the mechanical properties of 5-mol% yttria partially stabilized zirconia (5Y-PSZ) materials, designed for 3D printing or milling. Three 5Y-PSZ materials were investigated: printed zirconia (PZ) and two milled zirconia materials, VITA-YZ-XT (MZ-1) and Cercon xt (MZ-2). PZ samples were made from a novel ceramic suspension via digital light processing and divided into three subgroups: PZ-HN-ZD (horizontal nesting, printed with Zipro-D Dental), PZ-VN-Z (vertical nesting, printed with Zipro-D Dental) and PZ-VN-Z (vertical nesting, printed with Zipro Dental). Key outcomes included biaxial flexural strength (ISO 6872) and Vickers hardness (*n* ≥ 23 samples/subgroup). Microstructure and grain size were analyzed using light and scanning electron microscopy. Printed specimens exhibited biaxial flexural strengths of 1059 ± 178 MPa (PZ-HN-ZD), 797 ± 135 MPa (PZ-VN-ZD), and 793 ± 75 MPa (PZ-VN-Z). Milled samples showed strengths of 745 ± 96 MPa (MZ-1) and 928 ± 87 MPa (MZ-2). Significant differences (α = 0.05) were observed, except between vertically printed groups and MZ-1. Vickers hardness was highest for PZ-VN-Z (HV0.5 = 1590 ± 24), followed by MZ-1 (HV0.5 = 1577 ± 9) and MZ-2 (HV0.5 = 1524 ± 4), with significant differences, except between PZ and MZ-1. PZ samples had the smallest grain size (0.744 ± 0.024 µm) compared to MZ-1 (0.820 ± 0.042 µm) and MZ-2 (1.023 ± 0.081 µm). All materials met ISO 6872 standards for crowns and three-unit prostheses in posterior regions.

## 1. Introduction

Zirconia is becoming increasingly popular in dental restorations because of its favorable properties and broad indications [1]. Zirconia materials have become more translucent, expanding their use to monolithic restorations with good aesthetics [2,3]. Modifying the yttria content of zirconia to >3 mol% increases translucency and partially stabilizes the cubic phase; however, this advantage comes at the cost of reduced strength in 5 mol% yttria partially stabilized zirconia (5Y-PSZ) materials, although these modified zirconia materials are still stronger than other materials such as lithium disilicate glass ceramics [3,4]. Because of this reduced strength, restorations made from 5Y-PSZ are usually limited to single crowns or small fixed dental prostheses with a safety reserve.

Zirconia restorations are typically fabricated via a milling approach using computer-aided manufacturing (CAM) following a computer-aided design (CAD). Some years ago, additive fabrication approaches, including zirconia 3D printing, were introduced by the industry. Potential advantages of 3D printing over milling include more freedom in CAD design, no tooling stress and less material waste [5,6]. Thus, additive manufacturing of zirconia was rated a future-oriented technology [6,7]. Ongoing improvements in this technology have optimized the fit of 3D-printed restorations made from both zirconia [8] and lithium disilicate [9]. In addition, the flexural strength [10,11,12] and fracture resistance of 3D-printed restorations, including posterior occlusal veneers [13] or crowns [14], have been reported to be comparable with those of milled restorations. These findings make the clinical use of 3D-printed restorations interesting for translational research and practitioners.

However, a major drawback of printed 3Y-TZP is the increased risk of voids and flaws in the material, most likely attributed to the printing process itself, cleaning and the debinding of the printed green state. This reduces the strength of printed 3Y-TZP compared with milled 3Y-TZP [14,15]. The reduced strength of printed 3Y-TZP has also been linked to the nesting orientation in some 3D-printing systems [12,16]. Trueness and precision also seem to be affected by the nesting orientation [17] but with acceptable results for 0°, 45° and 90° tilt with respect to the printing orientation. To minimize these imperfections in zirconia materials and between the printing layers, some optimizations have been suggested, such as reducing the levelling speed and preparation schedules of the printing suspension. However, these approaches have not completely solved the problem. Furthermore, imperfections along the printing layers or in the material itself can also negatively affect optical properties.

Recently, a printable 5Y-PSZ material was developed, with the promise of a favorable translucency combined with strengths similar to that of the respective materials used with milling technology. However, the flexural strength of this new material has not been tested. Therefore, the aim of this study was to compare the biaxial flexural strengths of 3D-printed 5Y-PSZ with that of two commercial 5Y-TZP materials. Our first hypothesis is that milled 5Y-PSZ will have higher flexural strength and Vickers hardness than 3D-printed 5Y-PSZ. Our second hypothesis is that the use of different 3D printers will not affect the flexural strength. Our final hypothesis is that flexural strength will differ between vertically and horizontally nested 5Y-PSZ samples.

## 2. Materials and Methods

### 2.1. Materials

#### 2.1.1. Sampling

We investigated three 5Y-PSZ materials: printed zirconia (PZ) and two milled types of zirconia (MZ). The PZ samples were further divided into three subgroups, as described below. The chemical compositions of each material according to the manufacturer are listed in Table 1, and the sampling is further illustrated in Figure 1. Details of the three test materials are as follows:
PZ: these samples were fabricated from a novel ceramic slurry, INNI-CERA-T (AON Co. Ltd., Seoul, Republic of Korea), via digital light processing followed by a firing process consisting of debinding and sintering. The three subgroups were as follows:
a.PZ-HN-ZD: Horizontally nested (HN) PZ samples fabricated with the Zipro-D Dental printer (ZD; AON)b.PZ-VN-ZD: Vertically nested (VN) PZ samples fabricated with the Zipro-D Dental printerc.PZ-VN-Z: VN PZ samples printed using the Zipro Dental (Z; AON)
MZ-1: MZ samples fabricated from VITA YZ XT (VITA Zahnfabrik, Bad Säckingen, Germany)MZ-2: MZ samples fabricated from Cercon xt (Dentsply Sirona, Bensheim, Germany)
jfb-16-00036-t001_Table 1Table 1Chemical composition and flexural strength of the investigated materials according to the manufacturers.MaterialComponent [Weight %]Flexural Strength [MPa]ZrO_2_Y_2_O_3_HfO_2_Al_2_O_3_PZ88.58.22.70.1-MZ-186–918–101–30–1>600MZ-2>869<3<2750
Figure 1Sampling and study workflow.
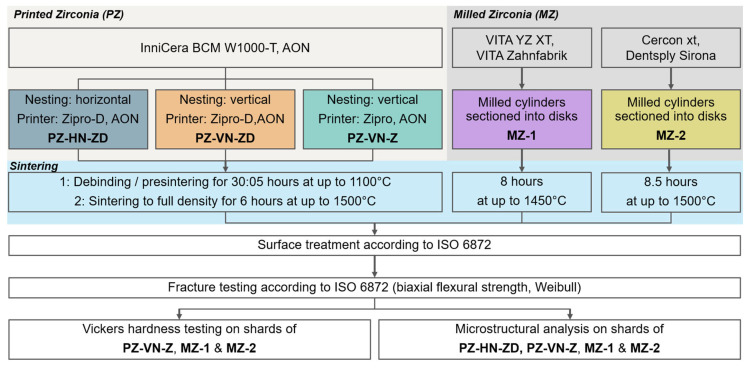



#### 2.1.2. Sample Size Determination

To calculate the sample size, two main effects on biaxial flexural strength were considered: the effect of nesting orientation within PZ samples and the effect of the manufacturing technique (3D printing vs. milling). Regarding nesting orientation, a previous study on printed 3Y-TZP zirconia [12] showed significant differences in biaxial flexural strength between the x-orientation (mean ± standard deviation [SD] 838 MPa ± 182 MPa, *n* = 20) and z-orientation (1107 MPa ± 144 MPa, *n* = 20). These values were used to calculate the Cohen’s d effect size according to Formulas 1 and 2. Employing G*Power [18] for a *t*-test comparing two independent means, with a resulting effect size of 1.64, significance level of 0.05, and power of 0.95, a sample size of 11 was determined for this comparison. Regarding the manufacturing technique, similar studies [11,12] compared the biaxial flexural strength of PZ (1107 MPa ± 144 MPa, *n* = 20) and MZ (1462 MPa ± 105 MPa, *n* = 20). With a resulting effect size of 2.82, significance level of 0.05, and a power of 0.95, a sample size of 5 was calculated for this comparison.(1)$$d=\frac{\bar{X_{1}}-\bar{X_{2}}}{s_{pooled}}$$(2)$$s_{pooled}=\sqrt{\frac{\left(n_{1}-1\right)s_{1}^{2}+\left(n_{2}-1\right)s_{2}^{2}}{n_{1}+n_{2}-2}}$$
with

${\bar{X}}_{i}$ = mean value of each group

*s_spooled_* = pooled standard deviation

*s_i_* = standard deviation of each group

*n_i_* = sample size of each group

To ensure robust statistics and account for potential variability, a higher number of samples was tested, with a minimum of 23 samples per (sub)group. This sample size was also sufficient for Weibull analyses (which require ≥20 samples per group).

#### 2.1.3. Sample Fabrication

Cylindrical specimens with a radius of 6.2 mm and a width of 1.6 mm were designed using CAD software (Geomagic DesignX; 3D Systems, Rock Hill, SC, USA) and exported in surface tessellation language files. The specimens designated for 3D printing were transferred to a slicing software (ZiproS Version 4.1 slicing software, AON), where they were horizontally or vertically nested (with respect to the building platform) and scaled. After adding support structures as recommended by the manufacturer, the samples were 3D-printed using zirconia slurry (InniCera BCM W1000-T, AON Co. Ltd., Seoul, Republic of Korea) and two different printers (ZD or Z), using a 405 nm light source to build the specimen layer by layer with a layer thickness of 50 µm. No differences regarding specific printing parameters between the fabrication with the two different printers were adjusted. After dismounting from the building platform, the samples were cleaned with isopropanol (purity ≥ 99.5%) using an airbrush system. Samples were debinded and presintered at up to 1100 °C for 30:05 h (ZIRFUR, AON Co. Ltd., Seoul, Republic of Korea) followed by final sintering process at up to 1500 °C for 6 h (SINTRA PRO/120zrf, Shenpaz Dental Ltd., Migdal HaEmek, Israel).

Samples designated for milling were fabricated as long cylinders (radius 6.2 mm after sintering) from commercial zirconia blanks using respective milling devices (MZ-1: PrograMill PM7, Ivoclar Vivadent AG, Schaan, Liechtenstein; MZ2: Cercon Brain Expert, Dentsply Sirona). In a second step, the long cylinders were sectioned into discs (width 1.6 mm after sintering) using a high-precision cutting machine (IsoMet High Speed Pro, Buehler, Lake Bluff, IL, USA). Samples were sintered according to the manufacturers’ guidelines: MZ-1 at up to 1450 °C for 8h (Programat S1 1600, Ivoclar Vivadent) and MZ-2 at up to 1500 °C for 8.5 h (Cercon heat plus, Dentsply Sirona).

Following ISO 6872 instructions, all samples were ground and polished (MD Piano diamond disks, #220, #500, #1200; Struers, Willich, Germany) in a semiautomatic grinding and polishing device (Tegramin25; Struers), giving a final sample width (b) of 1.2 mm ± 0.2 mm. To monitor the specimen dimensions, four width measurements (one at the center and three at the margin, spaced 120° apart) and two diameter measurements (at perpendicular spots) were taken using a digital micrometer screw (MicroMar 40 EWR, Mahr GmbH, Göttingen, Germany). Three measurements were taken at each spot and averaged to obtain a reliable value. All samples met ISO 6872 requirements (1.0 mm < b < 1.4 mm) and the exact dimensions were used to calculate the individual biaxial strength. Using digital microscopy (Smartzoom5, Zeiss, Jena, Germany), specimens were inspected for macroscopic flaws and voids and were excluded if any were observed.

### 2.2. Biaxial Flexural Strength Tests

The fracture tests were conducted using a universal testing device (Z005; Zwick/Roell, Ulm, Germany) with a cross-head speed of 1 mm/min. For biaxial strength calculations, a Poisson’s ratio of *ν* = 0.25 was assumed for all 5Y-PSZ materials. With the study setup displayed in Figure 2, sample geometry, and maximum force P, individual biaxial strengths can be calculated as follows:(3)$$\sigma =-0.2387\;P\frac{X-Y}{b^{2}}$$(4)$$X=\left(1+\nu \right)\ln{\left(r_{2}/r_{3}\right)}^{2}+\left[\left(1-\nu \right)/2\right]{\left(r_{2}/r_{3}\right)}^{2}$$(5)$$Y=\left(1+\nu \right)[1+\ln{\left(r_{1}/r_{3}\right)}^{2}]+\left(1-\nu \right){\left(r_{1}/r_{3}\right)}^{2}$$
with

ν = 0.25 (assumed according to ISO 6872 [19]);

*r_1_* = 5 mm, *r_2_* = 0.6 mm, *r_3_*: sample radius, *b*: sample thickness.
Figure 2Test setup for the biaxial flexural strength test in the universal testing machine. (**a**) An overview with situated sample. (**b**) The three-ball support of the lower part without specimen. The cylindrical indenter is located in the upper part and is lowered to the specimen while testing. To improve the overview, polyethylene foils were not placed between the sample and indenter and between balls and sample.
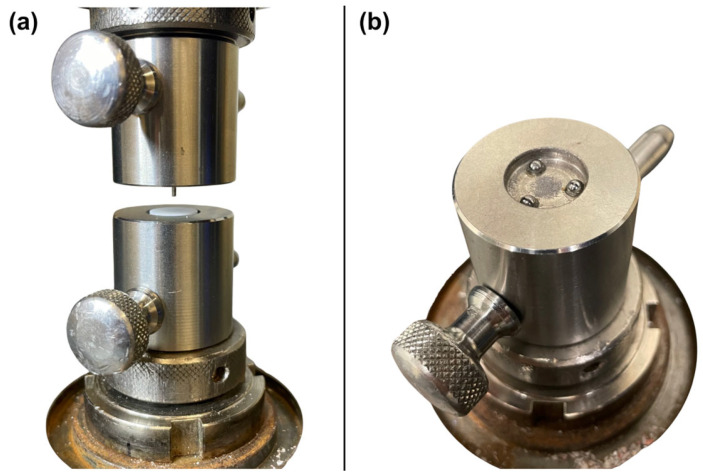



### 2.3. Vickers Hardness Measurement

Vickers hardness measurements were conducted with the shards from the biaxial flexural test samples using a Duramin-40 (Struers) semiautomatic hardness tester. A minimum of 23 samples per group were tested. For each sample, the largest shard was used, and three indentations with 1 mm spacing were made. Each measurement was performed with a load of F = 5 N (0.5-kg force) for 10 s, and the remaining indentation dimensions were measured at 50× magnification. The following equation was used to calculate the Vickers hardness (HV):(6)$$HV0.5=0.1891\;*\;\frac{F}{d^{2}}$$
with

*F* = applied load and *d* = mean diagonal length of the indentation

### 2.4. Imaging and Microstructural Analysis

First, shards from the biaxial flexural test specimen were visually inspected with a digital light microscope (Smartzoom 5, Carl Zeiss AG, Jena, Germany). For each material, four specimens were assessed at random, and two regions of interest were evaluated on each specimen. Prior to analysis, the specimens were ground and prepolished using #220, #500 and #1200 diamond disks and then polished using diamond suspensions and polishing cloths (first step Diapro Dac 3 µm/MD Dac and second step Diapro Nap B 1 µm/MD Nap, Struers). Thermal etching was performed for 30 min at 300 °C below the respective sintering temperature. Grain size was evaluated via SEM images (7500× magnification) by a single operator using the line intersection method (JSM 6510, Jeol, Akishima, Japan) at 7500×.

### 2.5. Statistical Analysis

All statistics were performed using SPSS version 29 (IBM; New York, NY, USA). Mean and SD values of the recorded flexural strengths and Vickers hardnesses were calculated for each group. For the biaxial flexural strength, Weibull distributions were fitted to the measured data, and Weibull parameters (characteristic strength: σ0, Weibull modulus: m) were calculated. Binomial confidence limits were also determined using median rank regression based on the beta-binominal distribution. The 5% and 95% ranks, taken from the standardized tables of DIN EN 61649 [20], were interpolated to obtain conservative beta-binominal intervals. These intervals were related to the Weibull line using the following equations:(7)$$t_{t0.95}=\sigma_{0}{\left[ln\left(\frac{1}{\left(1-F_{i(0.95)}\right)}\right)\right]}^{\left(\frac{1}{m}\right)}$$(8)$$t_{t0.05}=\sigma_{0}{\left[ln\left(\frac{1}{\left(1-F_{i(0.05)}\right)}\right)\right]}^{\left(\frac{1}{m}\right)}$$

These calculations determined the stress intervals to failure, considering the uncertainty due to limited sample size, and provided conservative estimates for the confidence limits.

Since individual test group data were normally distributed (Shapiro–Wilk tests) but individual group variances strongly differed between PZ and MZ materials, Welch ANOVA and Dunnet T3 pairwise post hoc tests were carried out to identify possible differences in biaxial flexural strength and Vickers hardness between groups. The level for statistical significance was α = 0.05.

Kruskal–Wallis and pairwise Mann–Whitney U-tests were performed to evaluate differences in grain size between PZ-HN-ZD, MZ-1 and MZ-2 samples.

## 3. Results

### Biaxial Flexural Strength

The biaxial flexural strengths of the different groups are shown in Table 2 and in Figure 3. PZ-HN-ZD had the highest biaxial flexural strength (mean ± SD, 1059 ± 178 MPa), and these specimens typically fractured into four or five shards. Samples with vertical nesting orientations had significantly lower mean biaxial strengths (PZ-VN-ZD, 797 ± 135 MPa and PZ-VN-Z, 793 ± 75 MPa; Welch ANOVA: *p* < 0.001, *p* < 0.001 for all pairwise tests), and these samples typically fractured into two or sometimes three pieces along the printing layer interfaces. There were no significant differences in biaxial flexural strength between different printers (*p* = 1.00) or between the vertically nested PZ samples and MZ-1 (*p* = 0.711, *p* = 0.457). MZ-2 had significantly higher biaxial flexural strength than MZ-1 (928 ± 87 MPa vs. 745 ± 96 MPa, *p* < 0.001), PZ-VN-ZD (*p* = 0.001), and PZ-VN-Z (*p* < 0.001).

Figure 4 shows that the Weibull distributions provided a good fit with the measured data. PZ-HN-ZD samples showed critical stress values of σc ≈ 580 MPa, 740 MPa, and 880 MPa. These values represent the critical stresses at which 5%, 50%, and 95% of the samples are expected to reach a low failure probability of P = 0.05 (5%), respectively, spanning a 300 MPa range. These critical stress values decreased to σc ≈ 440, 560, and 660 MPa (220 MPa range) for PZ-VN-ZD samples and to σc ≈ 560, 650, and 720 MPa (160 MPa range) for PZ-VN-Z samples. MZ-1 showed critical stress values of σc ≈ 460, 560, and 650 MPa (190 MPa range), and MZ-2 samples showed values of σc ≈ 690, 770, and 840 MPa (150 MPa range), respectively.

We also compared Vickers hardness between the different groups (Figure 5). Significant differences were found between PZ-VN-Z (1590 ± 24 HV0.5) and MZ-2 (1523.9 ± 3.9 HV0.5) (*p* = 0.036) and between MZ-1 (1577.0 ± 8.5 HV0.5) and MZ-2 (*p* < 0.001).

The main grain size was significantly different between the groups based on the microstructural analysis. PZ-VN-Z had the lowest mean grain size (0.744 ± 0.024 µm), followed by MZ-1 (0.820 ± 0.042 µm) and MZ-2 (1.023 ± 0.081 µm). Light and scanning electron microscopic images from representative 3D-printed samples revealed porosity that favorably aligned with the layer interfaces (Figure 6a). No such alignment was seen for the milled materials. Representative images for each material are depicted in Figure 6b–d, and the mean grain sizes are shown in Figure 7.

## 4. Discussion

The results of this study suggest that we have to partially reject our first hypothesis that MZ samples will have better biaxial flexural strength and Vickers hardness than PZ samples. Despite its microstructural defects, mainly in the form of pores, PZ-HN-ZD exhibited a higher mean biaxial flexural strength than MZ-1 and MZ-2, while PZ-VN-ZD and PZ-VN-Z showed similar strengths to MZ-1. Moreover, the mean Vickers hardness of PZ-VN-Z was higher or not significantly lower than that of MZ samples. This might be attributed to two factors. On the one hand, the slightly lower yttria content in the printed material compared to the milled materials results in a higher proportion of the tetragonal phase, which is accompanied with an overall smaller grain size. Together, these factors contribute to an enhanced flexural strength and Vickers hardness.

On the other hand, the microstructural flaws were predominantly aligned along the printing layers (Figure 6a), resulting in a weakened connection area. In general, such defects decrease the reliability of the material, which is reflected by the mostly lower Weibull moduli that we observed in PZ specimens. However, for PZ-HN-ZD samples, these weak areas between the layers are parallel to the effective normal stresses during the flexural test. Therefore, their effect on the specimen’s flexural strength is minor. Since hardness measurements are performed on a localized microscopic area of the sample, defects in the specimen that are relatively distant from the measurement site have no significant effect.

The results also showed that we have to accept our second hypothesis that there are no significant differences in flexural strengths between samples printed on the two different 3D printers. We also had to accept our third hypothesis that the nesting orientation significantly affects the biaxial flexural strength. The lower biaxial flexural strengths we observed in VN compared to HN PZ specimens (nesting orientation related to the building platform) were not surprising and were also seen for printed 3Y-TZP materials. As mentioned above, these lower strengths can be explained by the pores that align along the printing layers, which create an anisotropic structure and weaken connection areas. These weak connections are perpendicular to the effective normal stresses during biaxial flexural tests, leading to a lower flexural strength [12,16].

The lower flexural strength of MZ-1 compared to MZ-2 cannot be explained based on the results of the grain size determination. Therefore, additional factors like an increased number of defects or inhomogeneities in the microstructure may be the cause. Future investigations could clarify this matter.

Vickers hardness values show that the grain size affects the material properties. Specifically, the highest hardness was found in the material with the smallest grain size (PZ-V-Z), and the lowest hardness was found in the material with the biggest grain size (MZ-2), suggesting that hardness decreases with increasing grain size. It should be added that, in addition to the differences in chemical composition (see Table 1), the observed differences in grain size might also be due to the slight differences in final sintering time and temperature.

Our results regarding the biaxial flexural strength of MZ specimens are within the range of values provided by the manufacturers (see Table 1) and comparable, albeit partly higher, to findings in previous studies [21,22,23]. It is important to note that comparing our experimentally obtained values with those provided by the manufacturers or literature presents challenges. The values given by the respective manufacturer were determined using three-point bending tests. These tend to produce lower values, as beam-shaped samples are more prone to edge flaws and edge failures instead of showing a fracture that originates from an intrinsic flaw. During biaxial flexural strength testing, the highest tensile stress is located at the center of the surface opposite to the applied load. Therefore, edge flaws have no impact on the results [24,25,26]. Furthermore, even when using consistent test methods, previous studies have shown that variations in specimen preparation can significantly affect the results [27]. The results of the hardness measurements and the grain size evaluation of the milled materials are well within the range that is reported in the literature [21,22,28,29,30,31].

The printing process can impact the biaxial flexural strength. To evaluate this, we used two different printers from the same manufacturer to produce PZ samples while keeping the digital light processing technology and the slurry preparation the same. We observed no differences in biaxial flexural strength between the PZ samples depending on the printer used. Postprocessing of PZ samples, including cleaning and debinding, can also affect their fit, flexural strength, and reliability [8,32]. Therefore, we followed standardized and rather conservative protocols for the debinding and sintering protocol according to the manufacturer’s recommendations. It was beyond the scope of this study to consider factors that can increase flexural strength such as air particle abrasion, as we strictly tested according to ISO standards. A previous study using a different printer found that air particle abrasion increased the flexural strength and Weibull parameters [33], while another study found no effect in milled 3Y-TZP or 5Y-PSZ samples [34]. Limitations to this study need to be considered. First, Vickers hardness measurements were only performed on one PZ group. However, no significant differences between the PZ groups are to be expected, due to the above-discussed irrelevancy of defects that are distant from the microscopic measurement area. Secondly, an elaborate microstructural analysis of the two milled groups should be carried out in the future to identify possible defects that could explain their differing mechanical performance.

In the context of the intended clinical application in the oral environment, additional factors such as hydrothermal aging performance and biocompatibility are relevant factors for the applicability of the materials. Low-temperature degradation (LTD) is a frequently discussed topic regarding the aging behavior of zirconia as it can lead to the degradation of its mechanical properties [35,36,37]. Only limited studies investigating the LTD behavior of 3D-printed zirconia have been conducted until now, showing that even though the monocline phase content for the 3D-printed materials increased with aging, no significant decrease in flexural strength was observed [10,38,39]. Although these effects arise only after long periods of time, further research on this topic should be conducted in the future. More extensive research has been performed investigating the biocompatibility of 3D-printed zirconia, with predominantly positive outcomes [40,41].

## 5. Conclusions

Within the limitations of this laboratory study, we found that both printed and milled 5Y-PSZ samples had high biaxial flexural strengths, which justifies their use in single crowns and three-unit fixed partial dentures with a security reserve, according to ISO 6872. The PZ-HN-ZD and MZ-2 samples met the prerequisites for class 5 materials, which corresponds to four- and more-unit bridges. Further work could also investigate differences in translucency between PZ and MZ samples.

## Figures and Tables

**Figure 3 jfb-16-00036-f003:**
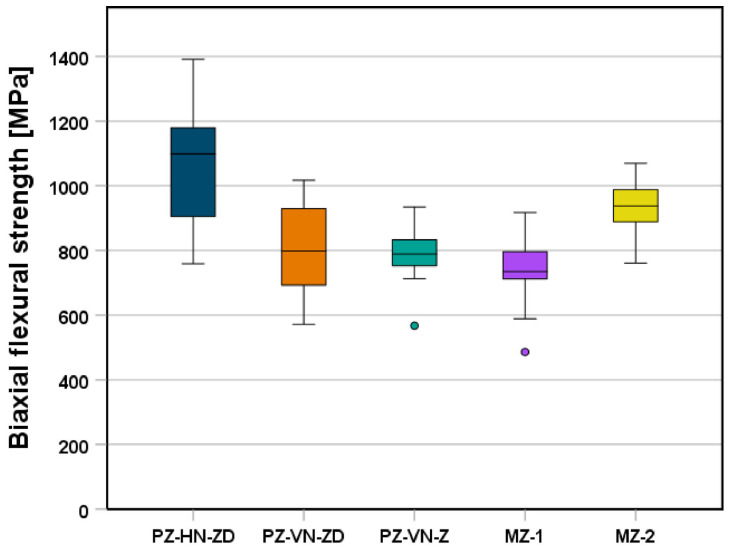
Boxplot diagram showing the biaxial flexural strength of the respective groups. PZ-HN-ZD = printed zirconia, horizontally nested, printed on Zipro-D Dental; PZ-VN-ZD = printed zirconia, vertically nested, printed on Zipro-D Dental; PZ-VN-Z = printed zirconia, vertically nested, printed on Zipro Dental; MZ-1 = VITA YZ XT; MZ-2 = Cercon xt. Circles indicate mild outliers that lie between 1.5 and 3 times the interquartile range (IQR) outside the quartiles.

**Figure 4 jfb-16-00036-f004:**
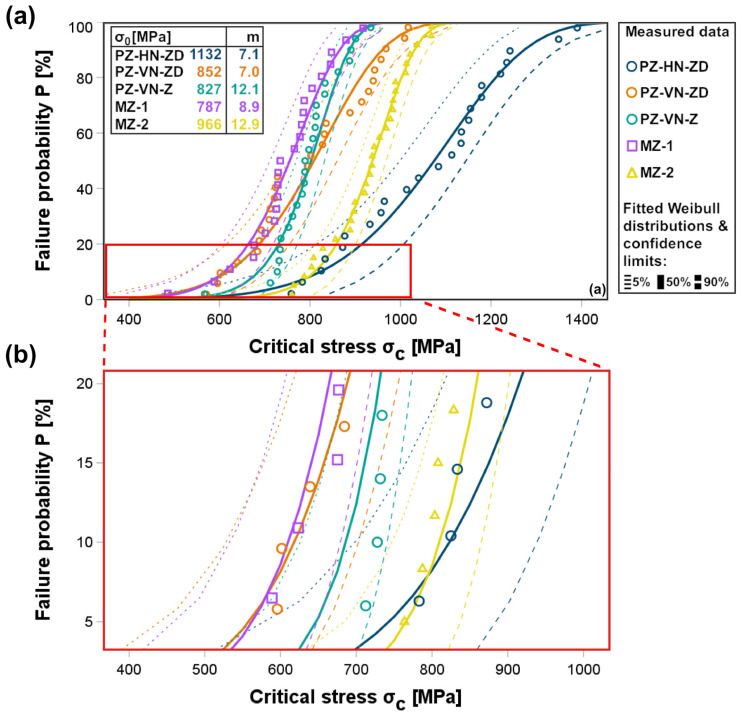
Weibull distributions for all groups. (**a**) The full range of distributions from 0 to 100% failure probability. (**b**) Amplification of the boxed area in (**a**), focusing on failure probabilities up to 20%. Symbols refer to the measured data and the graded lines refer to the fitted Weibull distributions with 5-, 50- and 90% confidence limits. Characteristic strength (σC) and Weibull modulus (m) for the respective group are shown in (**a**).

**Figure 5 jfb-16-00036-f005:**
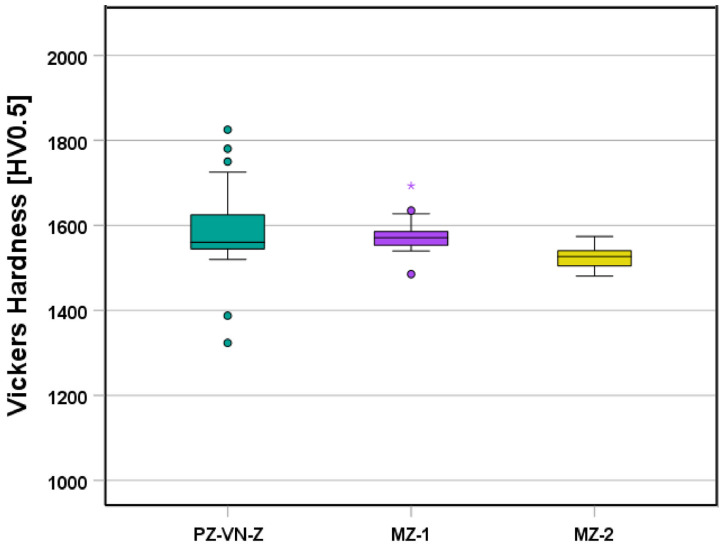
Boxplot diagram showing the Vickers hardness of the samples. Circles indicate mild outliers that lie between 1.5 and 3 times the interquartile range (IQR) outside the quartiles, while stars mark extreme outliers that are more than 3 times the IQR away from the quartiles.

**Figure 6 jfb-16-00036-f006:**
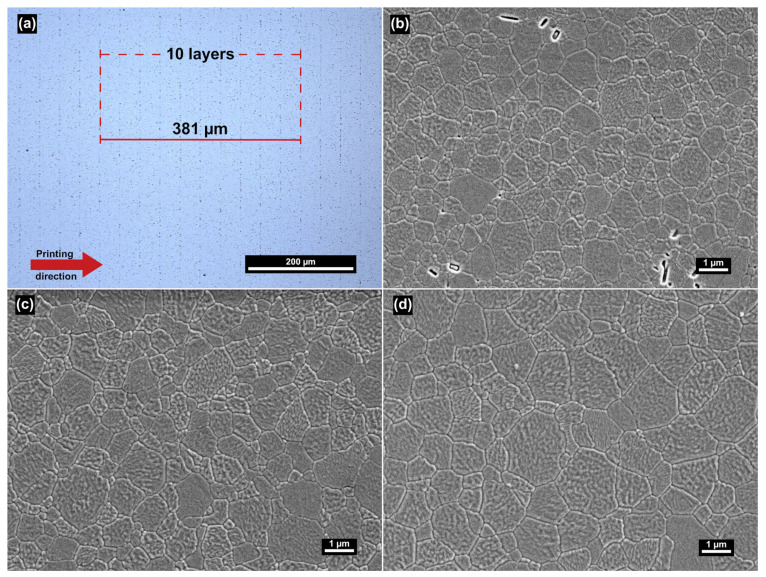
Light microscope (LM) and scanning electron microscope (SEM) images of the three materials. (**a**) LM image of PZ-VN-Z at 500 × magnification revealing a measured distance of 381 µm across ten printed layers, closely aligning with the calculated target value of 384.58 µm. This target was derived by dividing the layer height during printing (50 µm) by the scaling factor (1.3001), validating the conclusion of the pore-alignment along the print layers. (**b**–**d**) SEM images of (**b**) PZ-HN-ZD, (**c**) MZ-1 and (**d**) MZ-2 at 10,000× magnification.

**Figure 7 jfb-16-00036-f007:**
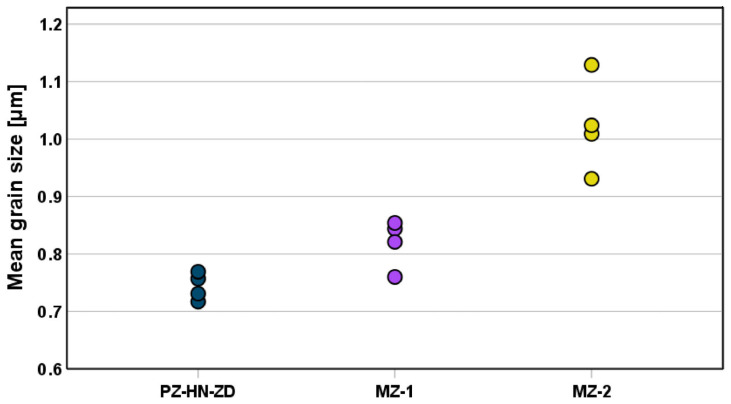
Scatter plot showing the mean grain size for each sample (four samples for each material).

**Table 2 jfb-16-00036-t002:** Biaxial flexural strength of the different test groups. Respective parameters are provided for normal distribution (mean value, standard deviation) as well as Weibull distribution (characteristic strength [σ0], Weibull modulus [m]). Different upper-case letters indicate significant differences between the test groups.

Material	Nesting Orientation	Printer	*n* [-]	Flexural Strength [MPa]		Weibull Parameters		Vickers Hardness [HV0.5]	
				Mean Value	SD	σ_0_ [MPa]	m [-]	Mean Value	SD
PZ	HN	ZD	24	1059 ^A^	178	1132	7.1	-	-
	VN	ZD	26	797 ^B^	135	852	7.0	-	-
	VN	Z	25	793 ^B^	75	827	12.1	1590 ^A^	24
MZ-1	-	-	23	745 ^B^	96	787	8.9	1577 ^A^	9
MZ-2	-	-	30	928 ^C^	87	966	12.9	1524 ^B^	4

## Data Availability

The original contributions presented in the study are included in the article, further inquiries can be directed to the corresponding author.

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
