# Peer review of "Biaxial Flexural Strength and Vickers Hardness of 3D-Printed and Milled 5Y Partially Stabilized Zirconia"

_jfb, 2025, doi:10.3390/jfb16010036_

Round 1

Reviewer 1 Report

Comments and Suggestions for Authors

The article compares the biaxial flexural strength and Vickers hardness of 3D-printed and milled 5Y-PSZ materials, which has strong clinical significance, but requires numerous revisions and supplementations.

  1. It is recommended to provide specific statistical power calculations, including the detailed process of determining the sample size and how the sample size for the current study was set based on the effect sizes observed in previous studies.
  2. The article mentions the use of two different 3D printers to manufacture PZ samples but does not detail the specific parameters of each technology, such as printing speed and exposure time. These parameters may affect the final performance of the material. The authors are advised to supplement these detailed information.
  3. There is a lack of analysis and discussion on the specific impact of microstructural defects on material performance. It is suggested to supplement the discussion on how microstructural defects (such as porosity, cracks, etc.) affect biaxial flexural strength and Vickers hardness.
  4. Supplement the standards and conditions of the tests.
  5. The materials studied are used in biomedicine. In the oral environment, how is their corrosion performance and biocompatibility? Reference relevant literature or tests to illustrate this.

Reviewer 2 Report

Comments and Suggestions for Authors

Overall the paper is interesting and the presented literature review is exhaustive.

I would just like to point out that the following two remarks need to be taken into account:

§  the authors should check for few spelling errors, typos, etc.

§ in the discussion section, since the grain size affects the materials properties and mainly the mechanical performances, the authors should try to give an explanation about the difference in the obtained grain size depending on the sample fabrication and operating parameters

Reviewer 3 Report

Comments and Suggestions for Authors

Congratulations! The article "Biaxial flexural strength and Vickers hardness of 3D-printed and milled 5Y partially stabilized zirconia" is a very interesting work that addresses an important and timely topic—differences between 3D-printed and milled zirconia restorations. From the perspective of a reviewer, I have only a few minor comments:

  • Why was HV not measured for the PZ HN and VN groups?
  • Figure 3 repeats data from Table 2—please consider whether this figure is truly necessary.
  • Please provide a more detailed description of the study's limitations in the discussion section.

Round 2

Reviewer 1 Report

Comments and Suggestions for Authors

It can be accepted.